# Antimicrobial Resistance, Pathogenic, and Molecular Characterization of *Escherichia coli* from Diarrheal Patients in South Korea

**DOI:** 10.3390/pathogens11040385

**Published:** 2022-03-23

**Authors:** Seong Bin Park, Yon Kyoung Park, Min Woo Ha, Kim D. Thompson, Tae Sung Jung

**Affiliations:** 1Coastal Research & Extension Center, Mississippi State University, Pascagoula, MS 39567, USA; sp1679@msstate.edu; 2Microbiology Division, Busan Institute of Health and Environment, Busan 616-100, Korea; akacia@korea.kr; 3Jeju Research Institute of Pharmaceutical Sciences, Jeju National University, Jeju 690-756, Korea; minuha@jejunu.ac.kr; 4Moredun Research Institute, Pentlands Science Park, Bush Loan, Penicuik EH26 0PZ, UK; kim.thompson@moredun.ac.kr; 5Laboratory of Aquatic Animal Diseases, Research Institute of Life Science, College of Veterinary Medicine, Gyeongsang National University, Jinju 660-701, Korea

**Keywords:** *E. coli*, pathotyping, phenotyping, genotyping, antimicrobial susceptibility test, South Korea

## Abstract

Diarrheal diseases due to foodborne *Escherichia coli* are the leading cause of illness in humans. Here, we performed pathogenic typing, molecular typing, and antimicrobial susceptibility tests on seventy-five isolates of *E. coli* isolated from stool samples of patients suffering from foodborne diseases in Busan, South Korea. All the isolates were identified as *E. coli* by both biochemical analysis (API 20E system) and matrix-assisted laser desorption/ionization-time of flight mass spectrometry (MALDI-TOF MS). The bacteria displayed entero-pathogenic *E. coli* (EPEC) (47.0%), entero-aggregative *E. coli* (EAEC) (33.3%), entero-toxigenic *E. coli* (ETEC) (6.6%), ETEC and EPEC (6.6%), EPEC and EAEC (4%), and ETEC and EAEC (2.7%) characteristics. The *E. coli* isolates were highly resistant to nalidixic acid (44.0%), tetracycline (41.3%), ampicillin (40%), ticarcillin (38.7%), and trimethoprim/sulfamethoxazole (34.7%); however, they were highly susceptible to imipenem (98.6%), cefotetan (98.6%), cefepime (94.6%), and chloramphenicol (94.6%). Although 52 strains (69.3%) showed resistance against at least 1 of the 16 antibiotics tested, 23 strains (30.7%) were susceptible to all the antibiotics. Nine different serotypes (O166, O8, O20, O25, O119, O159, O28ac, O127a, and O18), five genotypes (I to V, random-amplified polymorphic DNA), and four phenotypes (A to D, MALDI-TOF MS) were identified, showing the high level of heterogeneity between the *E. coli* isolates recovered from diarrheal patients in South Korea.

## 1. Introduction

Foodborne illnesses or foodborne diarrheal diseases affect the global population [1,2]. In the US alone, over 211 million cases of diarrheal disease are reported annually, resulting in more than 900,000 hospitalizations and 6000 deaths [2]. Foodborne diseases are categorized into foodborne infections or foodborne poisoning, and these are subdivided further according to the etiological agent involved, such as viruses, bacteria, fungi, parasites, or chemicals [3]. *Escherichia coli* is one of the most common bacterial agents of foodborne diarrheal disease [4]. *E. coli* is a Gram-negative, facultative anaerobic and rod-shaped bacterium, and is a common commensal bacterium in the large intestine of mammals [5]. However, some *E. coli* strains can cause serious food poisoning or infection by the ingestion of contaminated food [4]. Well-known strains of pathogenic *E. coli* include entero-toxigenic *E. coli* (ETEC), entero-invasive *E. coli* (EIEC), entero-pathogenic *E. coli* (EPEC), entero-aggregative *E. coli* (EAEC), Shiga toxin-producing *E. coli* (STEC), and entero-hemorrhagic *E. coli* (EHEC) based on the presence of virulence genes [6,7]. In South Korea during the period 2010–2018, pathogenic *E. coli* were responsible for the highest number of foodborne disease cases (1784–2754 cases) recorded, followed by Norovirus (739-1994 cases), non-typhoidal *Salmonella* (147–3516 cases), *Clostridium perfringens* (171–1689 cases) *Campylobacter* (380–831 cases), and *Staphylococcus aureus* (4–372 cases) [1].

To diagnose *E. coli*-related diseases, stool specimens are collected from patients displaying typical symptoms of *E. coli* infection, such as weakness, fever, and watery or bloody diarrhea, and cultured on selective agar or nutrient agar. Conventional biochemical tests (API or Vitek systems) and/or matrix-assisted laser desorption ionization time-of-flight mass spectrometry (MALDI-TOF MS) are then performed on isolated bacteria for their identification [8,9,10,11]. It is also essential to establish the correct antibiotic treatment for the patient, especially because of the emergence of multidrug resistant (MDR) *E. coli* [12]. MDR is defined as resistance acquired against at least one antimicrobial agent in three or more antibiotic categories. MDR *E. coli* are now recognized as one of the biggest challenges in public health worldwide [13].

Current molecular typing methods enable the differentiation between isolates of the same genus and species based on their phenotypic and genotypic characteristics [14]. Two representative molecular typing methods are used to determine the epidemiology of bacterial infections. One includes phenotyping-based methods such as O-serotyping, bacteriophage typing, and MALDI-TOF MS; while the other includes genotype-based methods such as restriction fragment length polymorphism (RFLP), amplified fragment length polymorphism (AFLP), random amplified polymorphic DNA (RAPD), pulse field gel electrophoresis (PFGE), and multi-locus sequence typing (MLST) [9,10,15,16,17,18,19,20,21].

In the present study, pathotyping (identification of pathogenic genes), phenotyping (O-serotyping and spectral analysis using MALDI-TOF MS), genotyping (clustal analysis using RAPDs), and antimicrobial susceptibility tests were conducted to investigate the characteristics of *E. coli* isolated from stool samples from foodborne diarrheal patients in Busan, South Korea.

## 2. Results

### 2.1. Identification and Pathogenic Properties of E. coli

A total of seventy-five bacterial isolates, obtained from diarrheal stool samples from foodborne diarrheal patients, were identified as *E coli* using API 20 E typing and MALDI-TOF MS using both bacterial colonies and extracted bacteria (log scores, ≥2.3; identification of species level). Thirty-six of the isolates (48%) were not assigned to any serotypes, while thirty-nine of the isolates (52%) belonged to only one *E. coli* O-serotype, i.e., O166 (8.0%, n = 6), O25 (5.3%, n = 4), O20 (5.3%, n = 4), O8 (5.3%, n = 4), O119 (4.0%, n = 3), O159 (4.0%, n = 3), O28ac (4.0%, n = 3), O127a (2.7%, n = 2), and O18 (2.7%, n = 2) (Table 1, Appendix A).

*St, eaeA, bfpA*, and *aggR* genes were amplified in all seventy-five *E. coli* isolates, while *lt, stx1, stx2,* and *ial* genes were not amplified (Table 1, Appendix A). All the isolates encoded at least one pathogenic gene, such as *eaeA* (35 isolates; EPEC pathotype), *aggR* (30 isolates; EAEC pathotype), *st* (12 isolates; ETEC pathotype), and *bfpA* (11 isolates; EPEC pathotype), but no genes for *E. coli* pathotypes EIEC, STEC, and EHEC were amplified. EPEC was the most prevalent pathotype (35 isolates; 47%), followed by pathotype EAEC (25 isolates; 33.3%), pathotype ETEC (5 isolates; 6.6%), pathotype ETEC and EPEC (5 isolates; 6.6%), pathotype EPEC and EAEC (3 isolates; 4%), and pathotype ETEC and EAEC (2 isolates; 2.7%). Of these, twelve isolates contained more than one virulence gene, while one isolate contained three virulence genes: *st, eaeA,* and *bfpA* (Table 1).

### 2.2. Antimicrobial Susceptibility 

Antimicrobial susceptibility was examined in all seventy-five *E.coli* isolates based on the standard CLSI protocol (Appendix A). Resistance was observed in all sixteen antibiotics examined (Table 2). The highest prevalence of resistance was to nalidixic acid with 44.0% of the isolates showing resistance (n = 33) followed by tetracycline at 41.3% (n = 31), ampicillin at 40.0% (n = 30), ticarcillin at 38.7% (n = 29), and trimethoprim/sulfamethoxazole at 34.7% (n = 26). Conversely, *E. coli* isolates were found to be most susceptible to imipenem at 98.6% (n = 74) and cefotetan at 98.6% (n = 74), followed by cefepime at 94.6% (n = 71), chloramphenicol at 94.6% (n = 71), cefazolin at 92% (n = 69), cefotaxime at 92% (n = 69), and amikacin at 92% (n = 69).

Twenty-three isolates (30.7%) were susceptible to all the antibiotics, while fifty-two isolates (69.3%) showed resistance against at least one of the sixteen antibiotics tested (Table 3). Thirty-four isolates (45.3%) showed MDR, with resistance to at least one in three or more of the antibiotic categories examined, of which two isolates were resistant to eleven antibiotics (Table 3).

### 2.3. Molecular Typing Using RAPD and MALDI-TOF MS

RAPD analysis was performed on the seventy-five isolates of *E. coli* using RAPD primer 5 of the Ready-To-Go-RAPD Analysis kit. This primer was chosen from the six primers provided in the kit because it generates clear and diverse amplification patterns. The same amplification patterns were observed in three independent experiments. RAPD analysis with primer 5 resulted in nine to nineteen clear bands, except for two isolates, which were not amplified (Figure 1). Five major clusters (I–V) were generated by primer 5, with a 76% similarity value between the clusters. Cluster I contained nine isolates (12%), and more than half of the isolates (n = 45) belonged to cluster II (60%, *p* < 0.05). Cluster III, IV, and V contained ten (13.3%), six (8%), and three (4%) isolates, respectively.

A dendrogram was generated based on the protein spectral fingerprints of the seventy-five *E.coli* isolates (Figure 2). The MALDI-TOF MS cluster analysis was performed using the integrated pattern-matching algorithm software exhibited, which divided the *E. coli* isolates into four major clusters (A to D) at a distance level of 500. Most isolates (68.0%, n = 51, *p* < 0.05) were classified into cluster D. Clusters A and B contained twelve (16.0%) and seven (9.3%) isolates, respectively. Five isolates were classified into cluster C (6.6%), which showed a close relative distance to cluster D at a distance level of 600.

The correlation analysis (Table 4) indicated that cluster D (68.0%, n = 51) from the RAPD analysis has a significant association with cluster III (60%, n = 45) from MALDI-TOF MS, since both clusters were the most dominant cluster of each molecular typing method (54.6%, n = 41).

## 3. Discussion

Characterization of pathogenic *E. coli* is performed using a variety of methods, including serotyping, genotyping, pathotyping, and phenotyping with various samples from human stools, dairy products, and meat from different countries [10,15,17,18,20,22,23]. Of these, the most well-known method for differentiating *E. coli* is serotyping based on 186 O antigens and 53 H antigens [6]. In general, an EHEC strain, O157:H7, is regarded as the major agent responsible for foodborne outbreaks since its shiga toxin (Stx) is essential for hemorrhagic colitis and hemolytic uremic syndrome in humans [24]. However, numerous studies have indicated that another fourteen non-O157 serotypes (O15, O26, O45, O55, O91, O103, O104, O111, O113, O118, O121, O128, O145, and O153) are becoming of public concern [24,25,26,27,28]. In this study, nine O-serotypes were identified in the *E. coli* collection, including O166, O8, O20, O25, O119, O159, O28ac, O127a, and O18, while no virulent STEC/EHEC O-serotypes were found. Interestingly, 48% of the isolates did not react with the 8 polyvalent and 43 monovalent serotyping sera that were used. Similarly, of the 101 *E. coli* isolated from chickens in South Korea in another study, only twenty-eight isolates (27.7%) could be serotyped, i.e., O1 (2%), O18 (3%), O20 (1%), O78 (19.7%), and O115 (2%) [23]. Similarly, of the 226 *E. coli* isolated from chickens in the Philippines, only five isolates (2.1%) could be O-serotyped [29]. In concurrence with these studies, our results showed the phenotyping of *E. coli* isolates, based on O-serotypes, to have a low discriminatory power because of the low number of isolates that could be assigned to a specific serotype. Conventional O-serotyping still plays a crucial role in confirming the pathogenic O-serotype in foodborne *E. coli* outbreaks, such as O157:H7 and other virulent O-serotypes.

Many research groups have focused on developing multiplex PCRs that are able to detect important pathogenic genes simultaneously because O-serotyping methods do not always indicate the exact pathogenicity of *E. coli* [10,15,16,17,19,21]. Based on these multiplex PCRs, some pathotypes have been found to be dominant in certain foods, hosts, and geographical regions. For example, of the 64 strains (22.9%) identified as diarrheagenic *E. coli* from a collection of 240 samples obtained from Tanzania, the most predominant pathotype was EAEC (14.6%), followed by EPEC (4.6%) and ETEC (3.6%), while EHEC and EIEC were not detected [30]. Similarly, in our study, EPEC (47.0%) was found to be the most abundant pathotype, followed by EAEC (33.3%) and ETEC (6.6%), while STEC, EHEC and EIEC were not found. Moreover, some strains were found to have genes encoding both pathogenicity islands, such as ETEC and EAEC (6.6%), ETEC and EPEC (4%), or EPEC and EAEC (2.7%), similar to a previous study that also demonstrated that several strains possessed two pathotypes [30]. It is reported that ETEC produces a heat-labile toxin (LT), with a sequence, antigenicity, and function similar to the cholera toxin, which is responsible for food poisoning in humans [31]. Some EPEC strains can colonize the small intestine and adhere to the mucosa causing effacement of microvilli, since a plasmid-encoding protein EPEC adherence factor (EAF) plays a key role in the pathogenesis of EPEC [32]. The distinctive characteristics of EAEC are the attachment of the *E. coli* to epithelial cells and the aggregation of the cells without penetration and replication [22,33]. In this study, our multiplex PCR showed that all seventy-five *E. coli* isolates possessed at least one pathogenic gene (EPEC, EAEC, and/or ETEC), which might be responsible for the disease seen in the diarrheal patients in South Korea. Moreover, all the isolates were divided into six groups (EPEC, EAEC, ETEC, ETEC and EAEC, ETEC and EPEC, and EPEC and EAEC) based on their pathogenic genes, which were further compared with two molecular typing analysis, RAPD (genotyping) and MALDI-TOF MS (phenotyping), to understand the correlation among the isolates.

To trace the source of foodborne illness and/or linking potential sources to the infection, studies using several molecular typing methods including RAPD, RFLP, PFGE, bacteriophage typing, and MALDI-TOF MS [9,15,16,17,18] have been used to classify the *E. coli* isolates involved. In this study, we conducted both a phenotyping analysis using mass spectrum profiles from MALDI-TOF MS and a genotyping analysis using RAPD fingerprinting. One study suggested that RAPD typing has great potential to identify the transmission of a particular bacterial species between different hosts and sources [34]. From the RAPD analysis, we detected genetic polymorphism among the 75 *E. coli* isolates, which were classified into five groups using primer 5 (I to V, Figure 1). Similarly, in a previous study analyzing 187 isolated from patients in an Iranian city from 2010 to 2016 using RAPD analysis, the *E. coli* isolates were associated with 32 different clusters, highlighting the amount of genetic diversity between the *E. coli* isolates from these patients [35]. On the other hand, several studies have indicated that MALDI-TOF MS analysis is able to discriminate between bacteria not only from various hosts and geographical origins, but also from the same host and region [36,37,38]. In this study, a dendrogram generated from MALDI-TOF MS indicated that *E. coli* strains showed heterogeneity in the phenotypes since they were divided into four clusters at the distance level of 500 (A to D, Figure 2). In a previous study, the dendrogram generated from MALDI-TOF MS failed to show any correlation between pathotypes, serotypes, and multilocus sequence types among the 136 *E.coli* strains examined [9]. Interestingly, they found that O157:H7 EHEC strains were relatively homogenous; however, the other pathotypes such as UPEC, EAEC, EIEC, DAEC, ETEC, and EPEC showed a variety of spectral patterns [9]. The current study demonstrated that pathotyping using pathogenic genes did not show any distinct distribution within the genotyping of RAPD clustering or within the phenotyping of MALDI-TOF MS clustering [39]. We concluded that the *E. coli* isolates recovered from human patients in the South Korea city of Busan were very heterogeneous with respect to pathotypes, genotypes, and phenotypes, based on the results of the O-serotyping, virulent gene amplification, and RAPD and MALDI-TOF MS analysis.

The present *E. coli* isolates were highly resistant to nalidixic acid, tetracycline, ampicillin, ticarcillin, and trimethoprim/sulfamethoxazole as well as highly susceptible to imipenem, cefotetan, cefepime, and chloramphenicol. They showed resistance to all sixteen antibiotics tested in this study. Drug-resistant *E. coli* has attracted great concern in relation to public health worldwide due to the abuse of antibiotics [40,41]. A previous study in South Africa demonstrated a high level of antibiotic resistance among *E. coli* isolates recovered from patients having foodborne diseases: cephalothin, 95%; streptomycin, 76%; ampicillin, 53%; amoxicillin-clavulanic acid, 5%; cefotaxime and kanamycin, 45%; sulfonamides and sulfa-trimethoprim, 21%; colistin, 16%; tetracycline, 13%; and ceftazidime and gentamicin, 8%. None of the isolates were resistant to chloramphenicol, ciprofloxacin, enrofloxacin, or nalidixic acid [42]. In a study from India, the resistance rates for *E. coli* isolates recovered from children were reported to be above 90% for cefotaxime, ceftazidime, cefepime, sulfamethoxazole, and co-trimoxazole. The isolates were highly susceptible for only two antibiotics: colistin and tigecycline [43]. However, in another study from Michigan in the United States, *E. coli* isolates (n = 353) from hospitalized patients during 2010–2014 were shown to have MDR to ampicillin (7.4%), trimethoprim/sulfamethoxazole (SXT) (4.0%), and ciprofloxacin (0.3%) [44], which are lower than the MDR levels mentioned above from South Africa [42], India [43], and Korea (in the present study). Taken together with the previous studies, our results suggest that *E. coli* isolated from patients from different countries may have different antibiotic susceptibilities, and therefore different strategies may be required to prevent and control the spread of MDR in foodborne pathogenic *E. coli*.

## 4. Materials and Methods

### 4.1. E. coli Isolates

Stool samples were collected from patients suffering from foodborne diarrheal diseases in Busan, South Korea, between 2014 and 2016, from which seventy-five isolates of *E. coli* were obtained. The stool samples were cultured on MacConkey (MAC; MERCK, Darmstadt, Germany) agar at 35 °C overnight, from which lactose-fermenting (pink) colonies were selected and cultured on triple sugar iron (TSI, Oxoid, Basingstoke, UK) slopes and tryptone soya agar (TSA, Oxoid, Basingstoke, UK). After Gram staining, the Gram-negative colonies showing gas production on TSI were cultured on TSA for identification using the API 20E typing system (BioMerieux, Durham, NC, USA).

### 4.2. O-Serotyping

O-serotypes were determined based on plate agglutination tests with O-antisera (DENKA SEIKEN, Tokyo, Japan). Briefly, bacterial colonies were suspended in 1 mL saline and incubated at 100 °C for 1 h followed by centrifugation at 900× *g* for 20 min. The pellet was resuspended in 0.5 mL saline and used as antigen for agglutination test. A total of 10 μL of antigenic suspension was mixed with the same volume of antisera or normal saline (as control for autoagglutination) on the slide glass. Agglutination was observed by tilting the glass slide backwards and forwards. If a specimen showed positive agglutination with the polyvalent serum, the O-serotype was then confirmed using the monovalent serum (shown in Appendix A) by observing agglutination with the bacteria within 1 min.

### 4.3. Characterization of Pathogenic Properties of E. coli by PCR

Genes associated with pathogenic properties of *E. coli* were determined by multiplex PCRs with minor modification [30,45,46,47,48,49]. The isolates were cultured in TSB at 35 °C for 24 h, and their DNA was extracted by boiling 50 μL of the culture with 450 μL of sterile distilled water for 20 min. Samples were then centrifuged at 14,000× *g* for 10 min, and supernatants were stored at −20 °C until used. Primer sets to amplify genes associated with pathogenic strains were used for multiplex PCR (Appendix A). Briefly, the mixture for multiplex PCRs included 2.5 units of i-StarMAX^TM^ DNA polymerase (INTRON, Seongnam-Si, Korea), 10 mM dNTP (2.5 mM each), 1 × PCR buffer, 2 μL of primer set (10 pmol), and 2 μL of bacterial DNA. The final volume was adjusted to 20 μL with distilled water. PCR was performed using T-100 thermal controller (Bio-Rad, Hercules, CA, USA) with the condition of 94 °C for 5 min, followed by 30 cycles of 94 °C for 1 min, 55 °C for 1 min, and 72 °C for 1 min, followed by a final extension at 72 °C for 5 min. The amplified PCR product was visualized on 1.5% TAE agarose gel containing 0.5 μg/mL ethidium bromide.

### 4.4. Antimicrobial Susceptibility Test

The antibiotic resistance of the isolates was determined using the disk diffusion method based on the standard procedure outlined by the Clinical and Laboratory Standards Institute (CLSI) [50,51]. *E. coli* isolates were spread on TSA and cultured at 35 °C for 24 h. A bacterial colony was suspended in 3 mL of Mueller–Hinton broth (MHB, Oxoid, Basingstoke, UK), and the concentration of the suspension was adjusted to 0.5 McFarland (BioMerieux, Durham, NC, USA). The bacterial suspension was evenly spread on Mueller–Hinton agar (Oxoid, Basingstoke, UK), and each antimicrobial disk (BD, Franklin Lakes, NJ, USA) was placed on the agar. The antimicrobial susceptibility of the *E. coli* isolates was assessed for 16 antimicrobial agents (BD, USA), including ampicillin (10 μg), amikacin (30 μg), chloramphenicol (30 μg), cephalothin (30 μg), ciprofloxacin (5 μg), cefotetan (30 μg), cefotaxime (30 μg), cefazolin (30 μg), cefepime (30 μg), gentamicin (10 μg), imipenem (10 μg), nalidixic acid (10 U), ampicillin/sulbactam (10 μg /10 μg), trimethoprim/sulfamethoxazole (1.25 μg /23.75 μg), tetracycline (30 μg), and ticarcillin (30 μg). After culturing the agar plates at 35 °C for 24 h, the diameter of the clear zones was determined using a digital caliper (Fisher Scientific, Hampton, NH, USA) to measure the growth inhibition of the isolates against the antimicrobial agents. The susceptibility of the isolates was determined according to CLSI (Appendix A). *E. coli* ATCC 25922 and *S. aureus* ATCC 29213 were used as quality control strains (data not shown).

### 4.5. Sample Preparation and Measurement of MALDI-TOF MS

Fresh bacterial colonies were applied directly onto the MSP 96 target polished steel plate (Bruker Daltonik, Bremen, Germany) and left to dry at room temperature. Thereafter, 1 μL of HCCA matrix solution (saturated solution of α-cyano-4-hydroxy-cinnamic acid in 50% acetonitrile and 2.5% trifluoroacetic acid) was added onto the samples and allowed to dry. For the standard extraction method, the bacterial colonies were suspended in 300 μL of distilled water and 900 μL absolute ethanol was added. After vortexing, the sample was centrifuged at 16,000× *g* for 2 min and the supernatant was discarded. The pellet was resuspended in 20 μL of 70% formic acid and mixed thoroughly by pipetting, followed by the addition of the same volume of acetonitrile. After centrifugation at 16,000× *g* for 2 min, 1 μL of supernatant was targeted onto the MSP 96 target polished steel plate and dried at room temperature before the addition of 1 μL of HCCA matrix solution to crystallize the sample.

The mass spectra were acquired using Microflex LT mass spectrometry (Bruker Daltonik, Bremen, Germany) under the control of Flexcontrol software (Version 3.0; Bruker Daltonik). Positive ions were extruded with an accelerating voltage of 20 kV, and the spectra were analyzed within a mass/charge (*m*/*z*) ratio of 2000 to 20,000 in the positive linear mode. Each spectrum was calibrated with a bacterial test standard (BTS 255343, Bruker Daltonik). For bacterial identification, the generated spectra were automatically matched with the reference library and scored using the integrated pattern-matching algorithm software (MALDI Biotyper RTC, Bruker Daltonik). A logarithmic score of 0 to 3 was assigned by MALDI Biotyper 3.0 according to spectra peak matching patterns. Scores of 0 to 1.699 indicated no reliable identification; 1.700 to 1.999 indicated a probable genus identification; 2.000 to 2.299 indicated a secure genus identification and probable species identification; and scores of 2.300 to 3.000 indicated highly probable species identification. In addition, MSP dendrograms were generated using MALDI Biotyper 3.0, and the distance level in the dendrograms was set at a maximal value of 1000 following the manufacturer’s recommendation.

### 4.6. Random Amplified Polymorphic DNA (RAPD) Analysis

Genomic DNA was extracted from *E. coli* isolates using the Accuprep genomic DNA extraction kit (Bioneer, Daejeon, Korea) following the manufacturer’s protocol. Random amplified polymorphic DNA (RAPD) analysis was performed with Ready-To-Go RAPD Analysis Beads (GE Healthcare, Piscataway, NJ, USA) following the manufacturer’s instructions. Briefly, each 25 μL reaction volume contained 10 ng of bacterial DNA and 25 pmol of a single RAPD primer. RAPD analysis primer 5 (5′-d[AACGCGCAAC]-3′) was applied to generate unique banding patterns. Amplification was performed using a PT-100 thermal cycler (MJ research, Watertown, MA, USA), and the protocol was 1 cycle at 95 °C for 5 min followed by 45 cycles of 95 °C for 1 min, 36 °C for 1 min, and 72 °C for 2 min. Visualization of PCR products was carried out as described above. The results were analyzed using Bionumerics software (Applied Maths, Austin, TX, USA) to generate a cluster analysis.

### 4.7. Statistical Analysis

Statistical analysis was performed using Graphpad Prism version 9.2.0. (GraphPad, San Diego, CA, USA). Significant differences were determined by the Chi-squared test or one-way ANOVA. The difference was considered statistically significant at *p* < 0.05.

## 5. Conclusions

The present study aimed to investigate the characteristics of *E. coli* isolated from stool samples of diarrheal patients in Busan, South Korea. The *E. coli* isolates demonstrated nine serotypes (O166, O8, O20, O25, O119, O159, O28ac, O127a, and O18), six pathotypes (EPEC, EAEC, ETEC, ETEC and EAEC, ETEC and EPEC, and EPEC and EAEC), five genotypes (I to V, RAPD), and four phenotypes (A to D, MALDI-TOF MS) with high antimicrobial resistances to sixteen antibiotics. This study helps to highlight the prevalence, phylogroups, and antimicrobial susceptibility of foodborne pathogenic *E. coli* isolated from South Korea.

## Figures and Tables

**Figure 1 pathogens-11-00385-f001:**
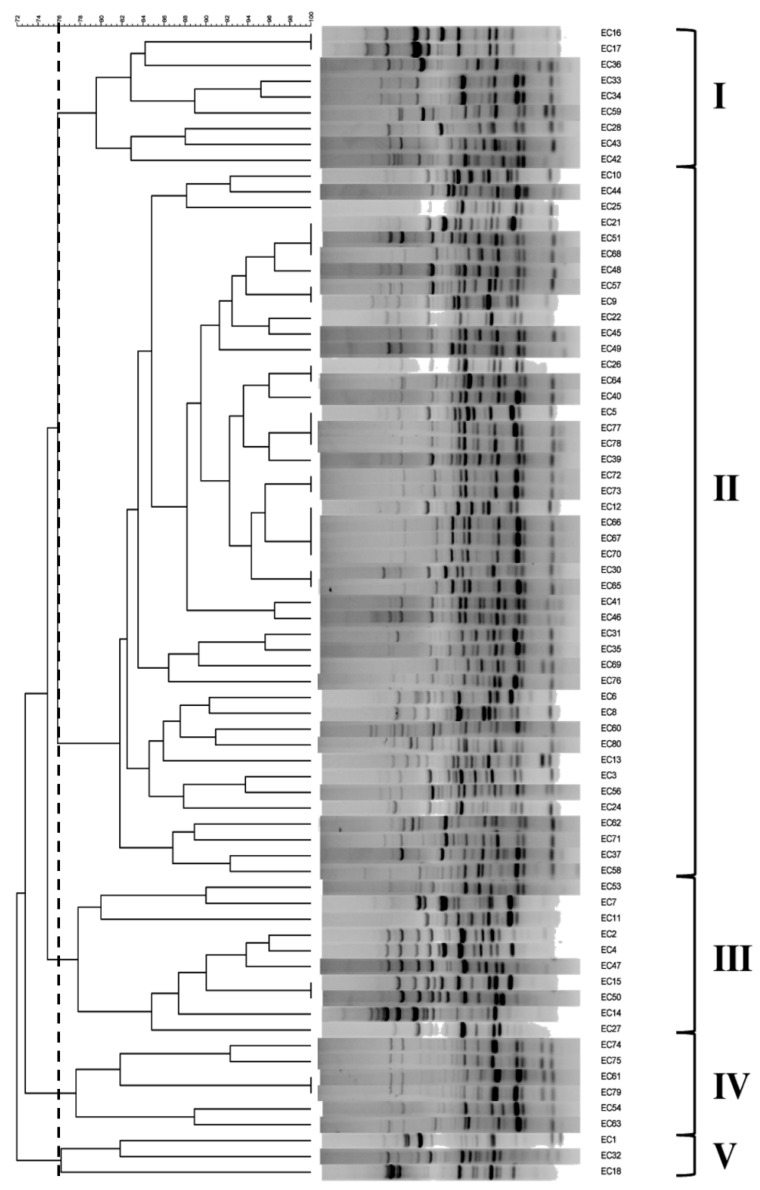
UPGMA cluster of RAPD band patterns for 75 *E. coli* isolates. The analysis was performed using RAPD primer 5 of the Ready-To-Go-RAPD Analysis kit. Five clusters were generated at 76% of RAPD profile similarity value using Bionumerics software.

**Figure 2 pathogens-11-00385-f002:**
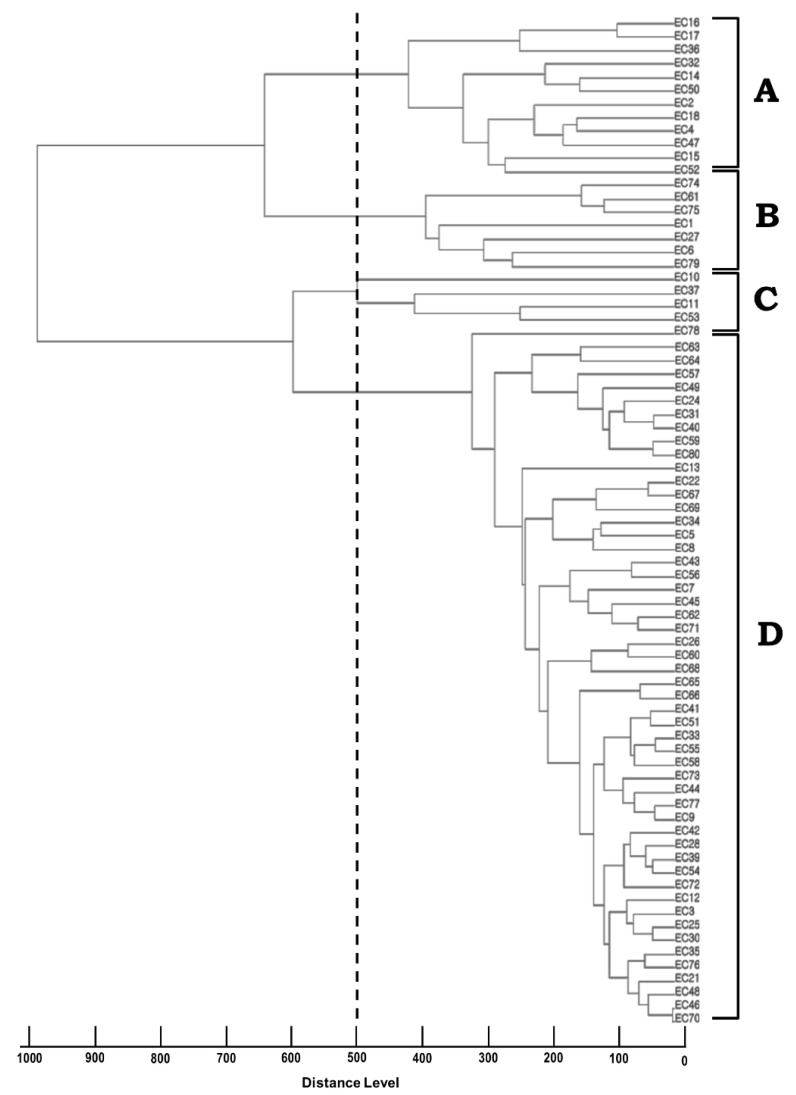
Cluster analysis of MSP dendrogram generated by MALDI-TOF MS for *E. coli* isolates. The dendrogram was generated using MALDI Biotyper 3.0, and the distance level in the dendrogram was set at a maximal value of 1000. The scale below the dendrogram signifies the relative distances. The five clusters were made at the distance level of 500.

**Table 1 pathogens-11-00385-t001:** O-serotyping and virulence gene distribution of *E. coli* isolates.

Pathotypes (%)	Virulence Genes	Total No. of Isolates	O Serotype (Total No. of Isolates)
ETEC (6.6)	*st*	5	O25(3), O159(1), UN(1)
ETEC and EAEC (2.7)	*st, aggR*	2	O159(1), UN(1)
ETEC and EPEC (6.6)	*st, eaeA, bfpA*	1	O8(1)
	*st, eaeA*	2	UN(2)
	*st, bfpA*	2	UN(2)
EPEC (47.0)	*eaeA*	27	O8(2), O18(2), O20(2), O55(1), O119(3), O125(1), O153(1), O166(2), O127a(2), O28ac(1), UN(10)
	*eaeA, bfpA*	2	UN(2)
	*bfpA*	6	O6(1), O153(1), UN(4)
EPEC and EAEC (4.0)	*eaeA, aggR*	3	O159(1), O169(1), UN(1)
EAEC (33.3)	*aggR*	25	O1(1), O8(1), O20(2), O25(1), O78(1), O166(4), O28ac(2), UN(13)

UN: unknown.

**Table 2 pathogens-11-00385-t002:** Antimicrobial resistance rate of *E. coli* isolates.

Antimicrobial (Abbreviations)	No. of Resistant Isolates (%)	No. of Intermediate Isolates (%)	No. of Susceptible Isolates (%)
Penicillins			
Ampicillin (AM)	30 (40.0)	0	45 (60)
Ticarcillin (TIC)	29 (38.7)	1 (1.3)	45 (60)
ß-Lactam/ß-Lactamase inhibitor combinations			
Ampicillin/Sulbactam (SAM)	12 (16.0)	8 (10.7)	55 (73.3)
Cephems			
Cefazolin (CZ)	6 (8.0)	0	69 (92)
Cefepime (FEB)	1 (1.3)	3 (4.0)	71 (94.6)
Cefotaxime (CTX)	4 (5.3)	2 (2.7)	69 (92)
Cefotetan (CTT)	1 (1.3)	0	74 (98.6)
Cephalothin (CF)	11 (14.7)	19 (25.3)	45 (60)
Phenicols			
Chloramphenicol (C)	4 (5.3)	0	71 (94.6)
Fluoroquinolones			
Ciprofloxacin (CIP)	10 (13.3)	4 (5.3)	61 (81.3)
Aminoglycosides			
Amikacin (AN)	5 (6.6)	1 (1.3)	69 (92)
Gentamicin (GM)	14 (18.7)	3 (4.0)	58 (77.3)
Carbapenems			
Imipenem (IPM)	1 (1.3)	0	74 (98.6)
Quinolones			
Nalidixic acid (NA)	33 (44.0)	1 (1.3)	41 (54.6)
Tetracyclines			
Tetracycline (TE)	31 (41.3)	0	44 (58.6)
Folate pathway inhibitors			
Trimethoprim/sulfamethoxazole (SXT)	26 (34.7)	2 (2.7)	47 (62.6)

**Table 3 pathogens-11-00385-t003:** Antimicrobial resistance patterns of 75 *E. coli* isolates.

No. of Antimicrobials	Resistance Patterns	No. of Isolates
0	-	23
1	SXT	1
	CZ	1
	TE	1
	NA	8
	TIC	2
	AM	1
2	TE, NA	1
	TE, TIC	1
	AN, TE	1
	AN, NA	1
3	AM, CF, SAM	1
	TE, NA, TIC	2
	CF, NA, TIC	1
	C, TE, NA	1
4	AM, SXT, NA, TIC	1
	AM, SXT, TE, SAM	1
	AM, NA, SAM, TIC	1
	AM, SXT, TE, TIC	3
	AM, SXT, TE, NA	1
5	AM, SXT, TE, NA, TIC	2
	AM, CF, C, TE, TIC	1
	GM, SXT, C, NA, TIC	1
	AM, CF, CIP, IPM, TE	1
	AM, GM, CTT, SAM, TIC	1
6	AM, CF, SXT, C, TE, TIC	1
	AM, CIP, SXT, TE, NA, SAM	1
	AM, GM, SXT, TE, NA, TIC	3
	AM, CZ, CF, SXT, TE, NA	1
	AM, CF, SXT, TE, NA, TIC	1
7	AM, GM, CIP, SXT, TE, NA, TIC	3
8	AM, GM, CIP, SXT, TE, NA, SAM, TIC	2
	AM, GM, AN, CTX, CIP, SXT, TE, SAM	1
	AM, CZ, CF, AN, SXT, NA, SAM, TIC	1
10	AM, CZ, CF, GM, AN, FEP, CTX, SXT, SAM, TIC	1
11	AM, CZ, CF, GM, CTX, CIP, SXT, TE, NA, SAM, TIC	2
Total	35 patterns	75 strains

SXT (Trimethoprim/Sulfamethoxazole), CZ (Cefazolin), TE (Tetracycline), NA (Nalidixic acid), TIC (Ticarcillin), AM (Ampicillin), AN (Amikacin), CF (Cephalothin), SAM (Ampicillin/Sulbactam), C (Chloramphenicol), CIP (Ciprofloxacin), IPM (Imipenem), GM (Gentamicin), CTT (Cefotetan), CTX (Cefotaxime), FEB (Cefepime).

**Table 4 pathogens-11-00385-t004:** Correlation analysis between clustal analysis by MALDI-TOF MS and phylogroup analysis by RAPD.

No. (%) of Strains	MALDI-TOF MS				
RAPD	A	B	C	D	Total N (%)
I	3 (4.0)	-	-	6 (8.0)	9 (12.0)
II	-	1 (1.3)	2 (2.7)	41 (54.6) *	44 (58.7) *
III	6 (8.0)	1 (1.3)	3 (4.0)	1 (1.3)	11 (14.7)
IV	-	4 (5.3)	-	2 (2.7)	6 (8.0)
V	2 (2.7)	1 (1.3)	-	-	3 (4.0)
None	1 (1.3)	-	-	1 (1.3)	2 (2.7)
Total N (%)	12 (16.0)	7 (9.3)	5 (6.7)	51 (68.0) *	75 (100)

* Statistical significance was assessed by Chi-squared test (*p* < 0.05).

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
