# Peer review of "Antimicrobial Resistance, Pathogenic, and Molecular Characterization of Escherichia coli from Diarrheal Patients in South Korea"

_pathogens, 2022, doi:10.3390/pathogens11040385_

Round 1

Reviewer 1 Report

In general, this is a very interesting and well-written manuscript. The authors isolated and characterized 75 E. coli isolates from the stool samples of patients with diarrheal diseases in South Korea. Using multiple approaches, the authors systematically analyzed the pathotypes, genotypes, and phenotypes of these isolates. More importantly, these isolates showed variable degree of antibiotic resistance. The analysis was designed and performed well, although the number of samples is relatively small and all obtained from the same city. I believe this study and the results presented here represent a great supplement of previous studies and may benefit the field a lot. Two minor points which are good to be addressed to move this manuscript to publishable level: (1) it is good to show the basic information of the patients and the association with the isolates; (2) was there non-pathogenic E. coli isolated from the stool samples and what’s the percentage?

Author Response

9-March-2022

 Pathogens-1628316-Antimicrobial resistant, pathogenic, and molecular characterization of Escherichia coli from diarrheal patients in South Korea

We appreciate the excellent efforts of the editor(s) and reviewers in handling/reviewing our manuscript and for providing the valuable comments to improve its quality. We have considered each comment very carefully and made the necessary corrections and revisions accordingly, and highlighted them in red in the revised manuscript. We believe our revisions have improved the quality of the manuscript and hope it is acceptable to you and the reviewers. 

Thank you for your continuous consideration.

Sincerely,

Tae Sung Jung

RESPONSE TO REVIEWER’S COMMENTS

Reviewer(s)' Comments to Author:

Reviewer #1:

In general, this is a very interesting and well-written manuscript. The authors isolated and characterized 75 E. coli isolates from the stool samples of patients with diarrheal diseases in South Korea. Using multiple approaches, the authors systematically analyzed the pathotypes, genotypes, and phenotypes of these isolates. More importantly, these isolates showed variable degree of antibiotic resistance. The analysis was designed and performed well, although the number of samples is relatively small and all obtained from the same city. I believe this study and the results presented here represent a great supplement of previous studies and may benefit the field a lot. Two minor points which are good to be addressed to move this manuscript to publishable level: (1) it is good to show the basic information of the patients and the association with the isolates; (2) was there non-pathogenic E. coli isolated from the stool samples and what’s the percentage?

Response: We appreciate Reviewer 1’s comment. The authors have very limit information on current E. coli isolates except the location and years because these E. coli strains were isolated from the stool samples from humans. Instead, the author put the ethical statement based on your comments (line 1070-1074); Institutional Review Board Statement: The Ethical Committee of the Gyeongsang National University confirmed that ethical review and approval were not needed since the study deals with a non-invasive diagnostic tool used according to standard indications in a specific subset of patients and clinical information for patients was collected anonymously (patient consent was not obtained such as name, address, code identifying an individual).

Reviewer 2 Report

Comments to the Author

The manuscript entitled " Antimicrobial-resistant, pathogenic, and molecular characterization of Escherichia coli from diarrheal patients in South Korea" represents a considerable amount of work. The following comments need to be addressed before the manuscript is suitable for publication in Pathogens Journal.

  • In the abstract, some important items are missing.
  • In Introduction: what is the background of Coli infection in diarrheic patients in South Korea? (reference)
  • The results section is not clearly written and is difficult to understand the data obtained, authors should revise this section for the form.
  • In material and methods,
  • please give details about the number of stool specimens examined
  • why didn’t authors statistically analyze their results?

Author Response

9-March-2022

 Pathogens-1628316-Antimicrobial resistant, pathogenic, and molecular characterization of Escherichia coli from diarrheal patients in South Korea

We appreciate the excellent efforts of the editor(s) and reviewers in handling/reviewing our manuscript and for providing the valuable comments to improve its quality. We have considered each comment very carefully and made the necessary corrections and revisions accordingly, and highlighted them in red in the revised manuscript. We believe our revisions have improved the quality of the manuscript and hope it is acceptable to you and the reviewers. 

Thank you for your continuous consideration.

Sincerely,

Tae Sung Jung

RESPONSE TO REVIEWER’S COMMENTS

Reviewer(s)' Comments to Author:

Reviewer #2:

The manuscript entitled " Antimicrobial-resistant, pathogenic, and molecular characterization of Escherichia coli from diarrheal patients in South Korea" represents a considerable amount of work. The following comments need to be addressed before the manuscript is suitable for publication in Pathogens Journal.

In the abstract, some important items are missing.

Response: We appreciate Reviewer 2’s recommendation. We have prepared a new abstract that contained more descriptions regarding pathotyping, genotyping, and phenotyping of E. coli strains. (Line 14-28).

In Introduction: what is the background of Coli infection in diarrheic patients in South Korea? (reference)

Response: Thank you for the comment. We have included the importance of foodborne pathogenic E.coli infection in South Korea with a reference (Line 75-79).

The results section is not clearly written and is difficult to understand the data obtained, authors should revise this section for the form.

Response: Thank you for the comment. We have made changes in the result section according to Reviewer 2’s comments.

In material and methods,

please give details about the number of stool specimens examined

Response: Thank you for the comment. The authors have very limit information on current E. coli isolates except the location and years because these E. coli strains were isolated from the stool samples from humans. Instead, the author put the ethical statement based on your comments (line 1070-1074); Institutional Review Board Statement: The Ethical Committee of the Gyeongsang National University confirmed that ethical review and approval were not needed since the study deals with a non-invasive diagnostic tool used according to standard indications in a specific subset of patients and clinical information for patients was collected anonymously (patient consent was not obtained such as name, address, code identifying an individual).

why didn’t authors statistically analyze their results?

Response: In the revised manuscript, the authors have conducted statistics according to reviewer’s suggestion (Highlighted copy: Line 383, 396, 400-402, Table 4 (Line 1044-1047), and 404-407).

Reviewer 3 Report

In the present study, pathotyping (identification of pathogenic genes), phenotyping (O58 serotyping and spectral analysis using MALDI-TOF MS), genotyping (clustal analysis using RAPD) and antimicrobial susceptibility tests were conducted to investigate the characteristics of E. coli isolated from stool specimens of foodborne diarrheal patients in Busan, South Korea.

The work is very interesting and important, however, there is some comments for improving the manuscript;

1- The introduction is not organized and not informative, you do not need to list the techniques that you did. You have to talk more about your topic, its economic importance. etc.

2- The number of the collected samples is very low. 

3- provide the ethical statement and the protocol approval.

4- Can you please provide a correlation between phenotypic and genotypic analysis.

5- provide the conclusion as a separate section

6- provide the statistical analysis for your results.

7- add the used statistical methods in separate section in materials and methods

Author Response

9-March-2022

 Pathogens-1628316-Antimicrobial resistant, pathogenic, and molecular characterization of Escherichia coli from diarrheal patients in South Korea

We appreciate the excellent efforts of the editor(s) and reviewers in handling/reviewing our manuscript and for providing the valuable comments to improve its quality. We have considered each comment very carefully and made the necessary corrections and revisions accordingly, and highlighted them in red in the revised manuscript. We believe our revisions have improved the quality of the manuscript and hope it is acceptable to you and the reviewers. 

Thank you for your continuous consideration.

Sincerely,

Tae Sung Jung

RESPONSE TO REVIEWER’S COMMENTS

Reviewer(s)' Comments to Author:

Reviewer #3:

In the present study, pathotyping (identification of pathogenic genes), phenotyping (O58 serotyping and spectral analysis using MALDI-TOF MS), genotyping (clustal analysis using RAPD) and antimicrobial susceptibility tests were conducted to investigate the characteristics of E. coli isolated from stool specimens of foodborne diarrheal patients in Busan, South Korea.

The work is very interesting and important, however, there is some comments for improving the manuscript;

1- The introduction is not organized and not informative, you do not need to list the techniques that you did. You have to talk more about your topic, its economic importance. etc.

Response: We appreciate Reviewer 3’s recommendation. Some important information in terms of the foodborne illness (Line 63-68) and the background of E. coli infection in diarrheic patients in South Korea (Line 75-79) have included in the revised manuscript according to the reviewer’s comments.

2- The number of the collected samples is very low. 

Response: In this study, seventy-five of E. coli strains were used to show the heterogeneity in pathotyping, genotyping, serotyping, and phenotyping with high resistances to antibiotics. A study published from South Africa used 38 E. coli strains to show the profiling of antimicrobial resistance and virulence [1]. A similar study from Qatar showed the monitoring data using 76 E. coli strains isolated from children [2]. Although the number of samples may not be sufficient, the authors believe our data may allow the monitoring of E. coli strains isolated from diarrheal patients in a city of Korea.

  1. Karama, M.; Cenci-Goga, B.T.; Malahlela, M.; Smith, A.M.; Keddy, K.H.; El-Ashram, S.; Kabiru, L.M.; Kalake, A. Virulence characteristics and antimicrobial resistance profiles of shiga toxin-producing Escherichia coli isolates from humans in South Africa: 2006–2013. Toxins 2019, 11, 424.
  2. Eltai, N.O.; Al Thani, A.A.; Al Hadidi, S.H.; Al Ansari, K.; Yassine, H.M. Antibiotic resistance and virulence patterns of pathogenic Escherichia coli strains associated with acute gastroenteritis among children in Qatar. BMC microbiology 2020, 20, 1-12.

3- provide the ethical statement and the protocol approval.

Response: We appreciate Reviewer 3’s comment. The authors have put the ethical statement based on your comments (line 1070-1074); Institutional Review Board Statement: The Ethical Committee of the Gyeongsang National University confirmed that ethical review and approval were not needed since the study deals with a non-invasive diagnostic tool used according to standard indications in a specific subset of patients and clinical information for patients was collected anonymously (patient consent was not obtained such as name, address, code identifying an individual).

4- Can you please provide a correlation between phenotypic and genotypic analysis.

Response: Thanks for your suggestion. We have statistically analyzed to show the correlation between phenotyping using MALDI-TOF MS and genotyping using RADP in the revised manuscript (Line 400-402, Table 4 (Line 415-417), and 1044-1047).

5- provide the conclusion as a separate section

Response: Thanks. The change has been made in the revised manuscript according to the reviewer’s comments (Line 1048-1055).

6- provide the statistical analysis for your results.

Response: In the revised manuscript, the authors have conducted statistics according to reviewer’s suggestion (Highlighted copy: Line 383, 396, 400-402, Table 4 (Line 1044-1047), and 404-407).

7- add the used statistical methods in separate section in materials and methods

Response: Thanks. According to the Reviewer’s suggestion, we have included the statistics in the separate section in materials and methods (Line 1044-1047).

Round 2

Reviewer 3 Report

The authors addressed all my comments.  However,  the manuscript still need English proof

Author Response

Reviewer(s)' Comments to Author:

Reviewer #3: The authors addressed all my comments.  However,  the manuscript still need English proof.

Response: We appreciate the excellent efforts of the reviewer 3 in handling our manuscript to improve its quality. We believe our revision have improved the quality of the manuscript and hope it is acceptable to the reviewer 3. Thank you.